# YOLO-Based Deep Learning Model for Pressure Ulcer Detection and Classification

**DOI:** 10.3390/healthcare11091222

**Published:** 2023-04-25

**Authors:** Bader Aldughayfiq, Farzeen Ashfaq, N. Z. Jhanjhi, Mamoona Humayun

**Affiliations:** 1Department of Information Systems, College of Computer and Information Sciences, Jouf University, Sakaka 72388, Saudi Arabia; bmaldughayfiq@ju.edu.sa; 2School of Computer Science, SCS, Taylor’s University, Subang Jaya 47500, Malaysia; farzeen.ashfaq@sd.taylors.edu.my (F.A.); noorzaman.jhanjhi@taylors.edu.my (N.Z.J.)

**Keywords:** classification of pressure ulcers, deep learning, object detection, YOLOv5

## Abstract

Pressure ulcers are significant healthcare concerns affecting millions of people worldwide, particularly those with limited mobility. Early detection and classification of pressure ulcers are crucial in preventing their progression and reducing associated morbidity and mortality. In this work, we present a novel approach that uses YOLOv5, an advanced and robust object detection model, to detect and classify pressure ulcers into four stages and non-pressure ulcers. We also utilize data augmentation techniques to expand our dataset and strengthen the resilience of our model. Our approach shows promising results, achieving an overall mean average precision of 76.9% and class-specific mAP50 values ranging from 66% to 99.5%. Compared to previous studies that primarily utilize CNN-based algorithms, our approach provides a more efficient and accurate solution for the detection and classification of pressure ulcers. The successful implementation of our approach has the potential to improve the early detection and treatment of pressure ulcers, resulting in better patient outcomes and reduced healthcare costs.

## 1. Introduction

Pressure ulcers, sometimes known as bed sores, are small, localized wounds to the skin and underlying tissues brought on by persistent pressure or friction against the skin [1]. They are serious health concerns, especially for older individuals and those with disabilities who spend a lot of time in bed or sitting down. However, newborn and pediatric children are also susceptible to pressure ulcers, albeit at a lower level [2]. Pressure ulcers are becoming more prevalent, particularly among elderly individuals who are more susceptible due to decreased mobility, sensory perception, and skin integrity [3,4]. Additionally, extended hospital stays in chronic illnesses, neurological conditions, complex organ failure, cancer, radiation treatments, and prolonged stays in the intensive care unit (ICU) [5], as well as long hospital stays during pandemics, such as COVID-19 [6,7,8], increase the risk of developing pressure ulcers. Moreover, the prevalence of chronic diseases, such as obesity, diabetes, and cardiovascular disease, is rising. Many of these diseases also increase the risk of developing pressure ulcers [9,10]. Sedentary lifestyles have led to more pressure ulcers, especially among older adults and those with disabilities [11,12,13,14]. The most commonly affected areas include the buttocks, head, shoulders, sacrum, coccyx, elbows, heels, hips, and ears. Figure 1 shows some of the areas where pressure ulcers may likely develop.

Patients who develop pressure ulcers may experience considerable effects, including pain, discomfort, and reduced mobility. In severe cases, they can lead to serious infections, hospitalization, and even death [15]. For the effective care and treatment of pressure ulcers, early and correct identification is essential [16]. However, classifying pressure ulcers into different stages can be challenging, especially for inexperienced healthcare providers [17,18]. As a result, there is a need for a more efficient and accurate method for classifying pressure ulcers that can improve patient outcomes and reduce healthcare costs. The current methods for classifying pressure ulcers into stages typically involve a visual assessment of the wound and a consideration of its depth and extent. There are several different classification systems in use, but the most widely used is the National Pressure Ulcer Advisory Panel (NPUAP) staging system [1]. Table 1 summarizes the NPUAP staging system classification of pressure ulcers into four stages based on the severity of tissue damage, ranging from stage I (least severe) to stage IV (most severe). Figure 2 provides a visual representation of the different stages.

Doctors and nurses typically classify pressure ulcers through visual inspection and manual palpation. During the assessment, they examine the wound site, look for signs of redness, blanching, and tissue loss, and take note of any other signs of infection or tissue damage. The healthcare provider may also use a probe or another instrument to assess the depth of the wound and evaluate any underlying tissue damage [19]. Accurate pressure ulcer classification is crucial, but healthcare providers can misclassify them due to various reasons. These include a lack of training and experience, limited information, observer bias, and variability in wound appearance. Factors such as the patient’s skin color, age, and overall health can also influence the visual appearance of a pressure ulcer, making it difficult to classify the wound without additional information [20]. Pressure ulcer classification requires a comprehensive assessment, considering various factors [21]. Deep learning, a subtype of machine learning inspired by the human brain, has shown potential in image classification by automatically learning complex patterns [22]. Computer vision and deep learning techniques have been widely adopted in various domains, including medical imaging, security, and image classification [23,24,25,26]. These techniques can detect and classify pressure ulcers by identifying visual features, such as color intensity, texture, depth, border, undermining, and tunneling, enabling earlier treatment and prevention of complications. Table 2 outlines the detectable features of pressure ulcers at each stage, which can be utilized by computer vision techniques, such as convolutional neural networks (CNNs) for accurate and efficient detection and classification.

Hence, we conclude that pressure ulcers are serious medical conditions that require accurate and timely detection and classification. While current methods rely on manual inspection, there has been increasing interest in developing automated approaches that can provide more accurate and efficient results. Various computer vision techniques, such as CNNs and manual image processing, have been explored for this task. However, previous studies have limitations, such as small dataset sizes, reliance on limited feature extraction methods, and lack of generalizability. Therefore, a more comprehensive and accurate solution to pressure ulcer detection and classification is needed [27,28,29,30,31,32,33].

To address this need, we propose a YOLO-based model for the detection and classification of pressure ulcers in medical images. Our method leverages the benefits of deep learning, such as its ability to automatically learn features, and employs data augmentation techniques to improve the model’s performance. We also use a dataset compiled from multiple sources with manually labeled images, which provides a more comprehensive and diverse set of training examples. Our study aims to accurately and efficiently detect and classify pressure ulcers, potentially leading to better patient outcomes and reduced healthcare costs.

Therefore, our work highlights the importance of image processing, computer vision, machine learning, and deep learning in the context of pressure ulcer detection and classification. By using a more comprehensive dataset and a more advanced deep learning model, we aim to address some of the limitations of previous studies and provide a more accurate and efficient solution to this critical healthcare problem. Our main contributions are as follows. We:Developed a YOLO-based deep learning model for the detection and classification of pressure ulcers in medical images, which can accurately classify pressure ulcers into four stages based on severity.Created a new dataset with the stage-wise classification of pressure ulcers, as no such dataset was previously available, by manually labeling images with bounding boxes and polygonal regions of interest using YOLO.Added a large number of images of non-pressure ulcers, including surgical wounds and burns, to the dataset to make it more representative of real-world scenarios.Utilized data augmentation techniques, such as rotation and flipping, to increase the dataset size and improve the model’s performance.Compared the performance of the proposed model to other cutting-edge deep learning models as well as to conventional pressure ulcer detection and classification methods, indicating the model’s advantage in terms of accuracy and effectiveness.Demonstrated the potential of the proposed model to aid in the timely diagnosis, treatment, and prevention of pressure ulcers, potentially improving patient outcomes and reducing healthcare costs.Contributed to the field by providing a new dataset and an accurate and efficient solution for the detection and classification of pressure ulcers, which has the potential to advance the state-of-the-art in this important area of medical image analysis.

In Section 2, we will first provide a comprehensive literature review of existing methods for pressure ulcer detection and classification, highlighting their strengths and limitations. In Section 3, we will present the materials and methods used in our study, including details of the dataset and image labeling process, as well as a description of the YOLO-based deep learning model and data augmentation techniques employed. Section 4 will report and discuss the results of our training and evaluation, including a comparison of our model’s performance with that of previous studies. In Section 5, we will provide an in-depth analysis of the model’s strengths and limitations, and discuss potential directions for future research. Finally, in Section 6, we will conclude our work with a summary of our key findings and contributions, as well as a discussion of the potential impact of our model on clinical practice and patient outcomes.

## 2. Literature Review

In recent years, a significant amount of research has been devoted to the use of deep learning in medical imaging. The literature review for this study primarily focuses on deep learning techniques applied to various medical imaging tasks, including object detection, image segmentation, and image classification. Figure 3 illustrates a hierarchy of tasks and algorithms for analyzing medical images using deep learning.

As seen in Figure 3, one area of research has been the development of convolutional neural networks (CNNs) for object detection in medical images. Researchers have applied CNNs to detect various structures in medical images, such as tumors, organs, and blood vessels. For example, Salama et al. and Agnes et al. [34,35] used a CNN to detect breast tumors in mammography images, Gao et al. and Monkam et al. [36,37] used a CNN to detect lung nodules in CT scans, and Li et al. and Ting et al. [38,39] used a CNN to detect retinal abnormalities in eye images. These studies have demonstrated the potential of CNNs for accurate and efficient object detection in medical images. Another method for identifying and segmenting liver tumors in multi-phase CT images is the phase attention mask R-CNN proposed by [40]. This method selectively extracts features from each phase using an attention network for each scale and outperforms other methods for segmenting liver tumors in terms of segmentation accuracy. Several other notable studies have explored the application of RCNN, Faster RCNN, and Fast RCNN in the automatic detection and segmentation of medical images, including works by [41,42,43,44,45,46].

Deep learning is being increasingly used for image segmentation in medical imaging. Researchers have developed deep learning algorithms for segmenting different parts of the body in medical photographs, such as the brain in MRI images [47], the lungs in CT scans [48], and the retina in eye images [49].

In addition to object detection and image segmentation, researchers have employed deep learning to classify medical images. The main focus of this study has been on creating algorithms for classifying medical images into specified categories, such as normal and abnormal, benign and malignant. For instance, Esteva et al. [50] used an intelligent algorithm to categorize the images of skin lesions, Shu et al. [51] used such a method to categorize mammography images, and Dansana et al. [52] used a deep learning system to categorize images of chest X-rays into normal and abnormal. Hence, we found extensive literature studies that have demonstrated that deep learning can successfully and properly categorize diseases in medical images.

Since our research focus is also mainly on object detection and classification in pressure ulcer images, we outline several noteworthy publications that have used deep learning algorithms for detection and classification tasks in medical images in Table 3.

YOLO (you only look once) is a recent deep learning approach used for object detection in medical images [60]. Compared to traditional CNN, YOLO is designed to be fast and efficient, making it well-suited for real-time object detection in medical images. YOLO divides the image into a grid and predicts bounding boxes and class probabilities for each cell. For example, in the case of lung cancer screening, the bounding boxes would represent the location of the lung nodules, and the class probabilities would indicate whether the nodule is cancerous or not. This can aid radiologists in the early detection and diagnosis of lung cancer, resulting in better patient outcomes. In medical imaging, YOLO has been applied to the detection of various structures, including tumors, organs, and blood vessels. For example, researchers have used YOLO to detect lung nodules in CT scans and retinal abnormalities in eye images; Boonrod et al. [61] used YOLO to detect abnormal cervical vertebrae in X-ray images, achieving higher accuracy and a faster processing time compared to traditional object detection methods. Similarly, Wojaczek et al. [62] used YOLO to detect the location and shape of prostate cancer in magnetic resonance images. Compared to traditional CNNs, YOLO has been shown to be faster and more efficient, while still achieving similar or even better performance in object detection tasks. Furthermore, YOLO is designed to be easy to train and implement, making it accessible to researchers and practitioners who may have limited experience with deep learning [63,64,65].

Various studies have explored the use of automated image analysis, deep learning, vision, and machine learning techniques for pressure ulcer detection, classification, and segmentation. These studies have demonstrated the feasibility of recognizing complicated structures in biomedical images with high accuracy, including the use of convolutional neural networks for tissue classification and segmentation, as well as the simultaneous segmentation and classification of stage 1–4 pressure injuries. However, these studies have their limitations, including the need for labeled datasets and the requirement for the manual selection of parameters. The authors of [66] proposed a system that utilized a LiDAR sensor and deep learning models for automatically assessing pressure injuries, achieving satisfactory accuracy with U-Net outperforming Mask R-CNN; Zahia et al. [67] used CNNs for automatic tissue classification in pressure injuries, achieving an overall classification accuracy of 92.01%; Liu et al. [68] developed a system using deep learning algorithms to identify pressure ulcers and achieved high accuracy with the Inception-ResNet-v2 model; Fergus et al. [69] used a faster region-based convolutional neural network and a mobile platform to classify pressure ulcers; Swerdlow et al. [70] applied the Mask-R-CNN algorithm for simultaneous segmentation and classification of stage 1–4 pressure injuries; Elmogy et al. [71] proposed a tissue classification system for pressure ulcers using a 3D-CNN. Table 4 summarizes these studies.

In this study, YOLO is employed for the detection and classification of pressure ulcers into four stages, and non-pressure ulcer images are also considered. Augmentation techniques are used to enhance the quality and quantity of the available dataset. By collecting images from various sources, the study aims to overcome the limitations of previous studies and provide a more accurate and robust system for pressure ulcer detection and classification. This study’s results will be important for medical professionals and caregivers to make informed decisions on the prevention and treatment of pressure ulcers.

## 3. Methodology

### 3.1. Description of the YOLO Model

YOLOv5 is a deep neural network architecture used for object detection that was developed by Ultralytics [60,72,73]. Its performance improves previous versions by introducing a number of architectural improvements and optimizations. A detection head (DH) and a backbone network (BN) make up the YOLOv5 architecture. The BN is in charge of feature extraction, and the DH is responsible for the actual object detection and categorization.

The backbone network of YOLOv5 is based on a modified version of the EfficientNet architecture, which uses a compound scaling method to balance the model size and performance. The EfficientNet architecture was originally proposed by Tan and Le in 2019 and has since become a popular choice for many computer vision tasks.

The detection head of YOLOv5 is similar to that of YOLOv4, as shown in Figure 4, but with some important modifications. It uses spatial pyramid pooling (SPP) to capture features at different scales and a new anchor system that is better suited for detecting small objects.

Overall, YOLOv5 is designed to be fast, accurate, and easy to use. It achieves state-of-the-art performance on a number of object detection benchmarks while maintaining a small model size.

We selected YOLOv5s for pressure ulcer detection and classification due to its high accuracy in object detection, real-time performance, and flexibility in training on custom datasets. Additionally, YOLOv5 has been successfully applied in several other medical imaging tasks, including lung nodule detection and diabetic retinopathy detection, which suggests that it could be effective for pressure ulcer detection and classification as well.

### 3.2. Dataset Used for Training and Testing

In order to develop a highly accurate deep learning-based model for pressure ulcer classification into stages 1, 2, 3, and 4, as well as non-pressure ulcers, a diverse and comprehensive dataset was collected for both training and evaluation. This dataset was thoughtfully designed to include two distinct sets of pressure ulcer images, as well as images of other types, such as burns, abdominal wounds, and diabetic foot ulcers, all of which were sourced from the highly regarded Medetec image database [75].

To increase the size and diversity of the dataset, an additional 200 images from all 4 stages and non-pressure ulcers were collected from various online sources, including Google Images. To further enhance the dataset, advanced data augmentation techniques were applied, including image rotation, flipping, and resizing, to generate additional images from the existing data. The careful selection and augmentation of this diverse dataset will enable the deep learning model to be robust to variations in input data and to generalize well to new, unseen pressure ulcer images. Figure 5 illustrates the class distribution of the dataset after the inclusion of these additional images and data augmentation techniques.

The labeling process of the dataset was conducted in collaboration with a medical domain expert to ensure accuracy and consistency in determining the stages of pressure ulcers. The images were labeled with polygonal bounding boxes to precisely outline the areas of interest for the deep learning model.

## 4. Results

We conducted our experiments on a Colab GPU with YOLOv5 version 7.0-114-g3c0a6e6, Python version 3.8.10, and Torch version 1.13.1+cu116. We trained the model using the following hyperparameters: a learning rate (lr0) of 0.01, momentum of 0.937, weight decay of 0.0005, and batch size of 18. We used the stochastic gradient descent (SGD) optimizer for 500 epochs with patience of 100, and saved the best model weights.

Based on the YOLOv5s model we trained, we achieved good results in terms of the overall mAP and individual class performance. The model achieved an overall mAP50 of 0.769 and mAP50-95 of 0.398 on the validation set. This means that the model was able to accurately detect and classify pressure ulcers with a high degree of confidence.

Figure 6 shows the loss values for the box loss, object loss, and class loss at each epoch during the training process. The box loss represents the difference between the predicted and ground-truth bounding box coordinates, the object loss represents the confidence score for each object detected in an image, and the class loss represents the probability of each detected object belonging to a specific class.

The goal of training an object detection model is to minimize the total loss, which is a combination of box loss, object loss, and class loss. The loss values should exhibit a decreasing trend as the training progresses, indicating an improvement in the model’s ability to detect different stages of pressure ulcers in the images.

Moreover, from Figure 7, it appears that the precision, recall, and mean average precision (mAP) are all increasing with training epochs. This could indicate that the model improves over time and becomes more accurate at identifying the correct stages of pressure ulcers in the images.

From Table 5, it can be observed that the model attained high precision and recall scores for the non-pressure ulcer (NonPU) and Stage 1 categories, suggesting that the model accurately detected and classified these classes. However, the recall score for Stage 2 was relatively low at 0.164, indicating that the model may have missed some instances of this category, possibly due to the small number of images available for this class.

Stage 3 and Stage 4 of our model have shown promising results in detecting and classifying pressure ulcers. Our proposed model achieved a mean average precision (mAP50) of 0.749 and 0.729, respectively, indicating a high level of accuracy in identifying and localizing pressure ulcers in the images. In Stage 3, the model accurately identified 26 instances of pressure ulcers with a precision (P) of 0.659 and recall (R) of 0.692, demonstrating its ability to detect and classify pressure ulcers even in challenging image conditions. In Stage 4, the model identified 22 instances of pressure ulcers with a precision (P) of 0.615 and recall (R) of 0.864, indicating its capability to correctly identify and classify a significant number of pressure ulcers with high accuracy.

Overall, the results of our YOLOv5s model suggest that it performed well in accurately detecting and classifying pressure ulcers in the images we used for training and validation. Hence, we can infer that these results demonstrate the potential of the YOLOv5 model for detecting and classifying pressure ulcers in medical images, which could have significant implications for improving patient care and outcomes.

In addition to the metrics presented in Table 5, we also generated additional evaluation metrics to further analyze the performance of our YOLOv5 model. The precision confidence curve, recall confidence curve, precision–recall (PR) curves, F1 score curves, and confusion matrices can be found in Figure 7, Figure 8, Figure 9, Figure 10, and Figure 11, respectively. These evaluation metrics provide a more detailed understanding of the model’s ability to accurately detect and classify pressure ulcers in the images. The PR curves and F1 score curves show the trade-off between precision and recall for different decision thresholds, while the confusion matrices provide information on the number of true positives, true negatives, false positives, and false negatives for each class.

## 5. Discussion

While there have been numerous studies on pressure ulcer detection and classification using deep learning, YOLOv5 has not been commonly used in this domain. In fact, to the best of our knowledge, there are no studies that have employed YOLOv5 for this task. Only one notable study has used YOLOv4 for pressure ulcer detection and classification. Additionally, distinguishing between pressure and non-pressure ulcers using deep learning has not been widely explored. Hence, our work makes a novel contribution to the field by utilizing YOLOv5 for pressure ulcer classification and introducing a novel approach to simultaneously distinguish between four stages of pressure ulcers and non-pressure ulcer images, making it a significant contribution to the field. Furthermore, it is worth noting that for object detection tasks, accuracy alone may not be a sufficient metric, as it does not account for false positives and false negatives. Instead, mean average precision (mAP) is often used to evaluate the performance of object detection models. The mAP takes into account precision and recall at different levels of intersection over union (IoU) thresholds, and provides a more comprehensive evaluation of the model’s performance. In our study, we report a mAP50 of 0.769, which indicates that our model performs well in detecting and classifying pressure ulcers into different stages. In Table 6, we present a comparative analysis of our findings with three benchmark studies.

It is evident from the table that our model performs better than the benchmark models in terms of the quantity and variety of classes in the dataset employed. Although, Liu et al. [68] achieved higher recall and precision scores than our model, it is important to note that the two models were trained and evaluated on different tasks. While our model was trained to detect and classify pressure ulcers into five classes, whereas the CNN-based model was trained to classify images as either erythema or non-erythema, and to classify tissue as necrotic or non-necrotic. Therefore, it is not necessarily a fair comparison to directly compare the performance metrics of the two models. Hence, it is important to highlight that because the performance indicators employed in these research studies differ, it is challenging to directly compare the results. Our model outperformed [29,69], as demonstrated in the table, with superior metrics and a significantly higher mAP value. These results demonstrate the potential of our model for accurate pressure ulcer detection and classification. Although promising, further improvements can be made by fine-tuning the hyperparameters and optimizing the model to enhance its performance.

## 6. Conclusions

Our study demonstrates the effectiveness of a deep learning-based approach for the automated detection and classification of pressure ulcers. By utilizing the YOLOv5 model, we achieved high accuracy in identifying pressure ulcers of different stages and non-pressure ulcers. Our proposed model outperformed existing CNN-based methods, showcasing the superiority of YOLOv5 for this task.

We also developed a diverse dataset of pressure ulcers and non-pressure ulcers, which we augmented to enhance the model’s robustness to variations in input data. Our model achieved high precision and recall scores for non-pressure ulcers and Stage 1 pressure ulcers, which are easier to identify due to their distinct features. However, the recall score for Stage 2 pressure ulcers was relatively low due to the limited number of images available for this class. To improve the model’s performance, we recommend collecting more images of Stage 2 pressure ulcers.

Our study’s findings demonstrate the potential of deep learning-based systems for automated pressure ulcer detection and classification, offering a promising solution for earlier intervention and improved patient outcomes. This technology could also reduce the workload of healthcare professionals, allowing them to focus on other essential tasks. To further advance this field, we plan to investigate a tailored YOLOv5 model trained on a larger and more diverse dataset, including images from different populations, races, and age groups. Additionally, we will explore the use of transfer learning by leveraging pre-trained models on other medical imaging datasets.

## Figures and Tables

**Figure 1 healthcare-11-01222-f001:**
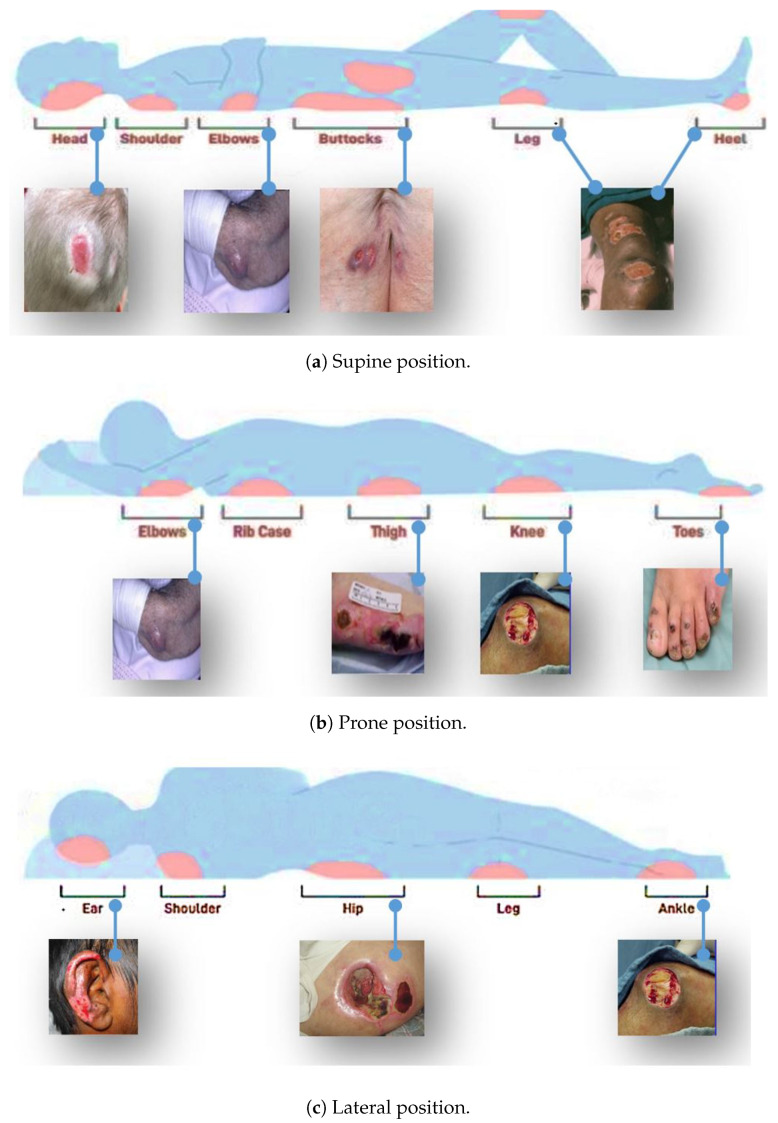
Positions and areas of the body at risk for pressure ulcers.

**Figure 2 healthcare-11-01222-f002:**
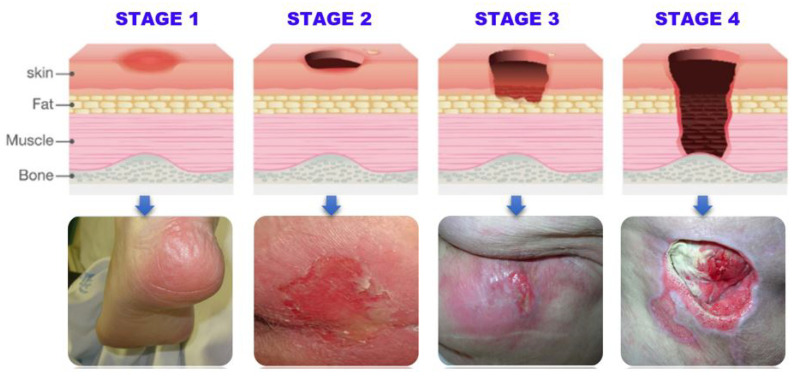
The Stages of Pressure Ulcer.

**Figure 3 healthcare-11-01222-f003:**
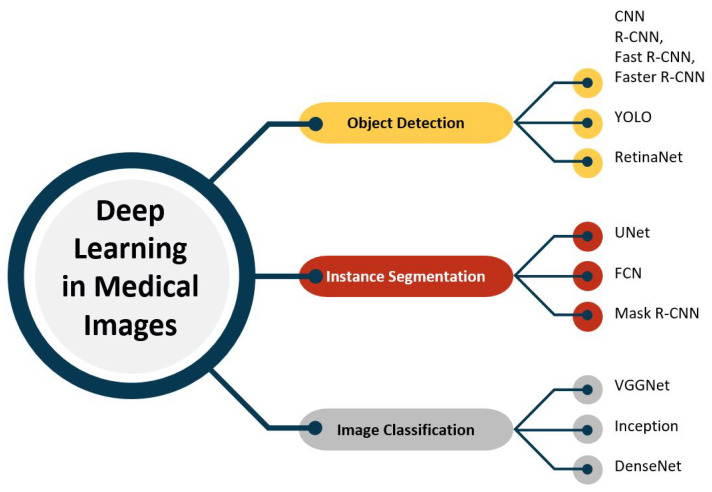
Applications of deep learning algorithms for medical images.

**Figure 4 healthcare-11-01222-f004:**
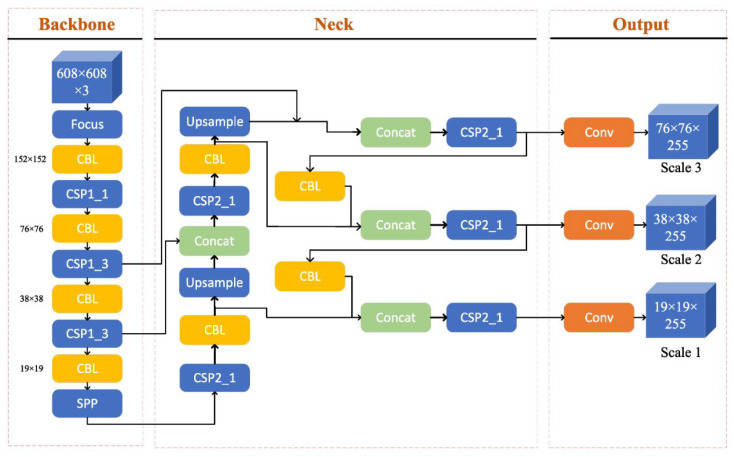
Architecture diagram for YOLOv5, adapted from [74]. YOLOv5 introduced a new architecture that includes a scaled YOLOv3 backbone and a novel neck design, which consists of SPP and PAN modules.

**Figure 5 healthcare-11-01222-f005:**
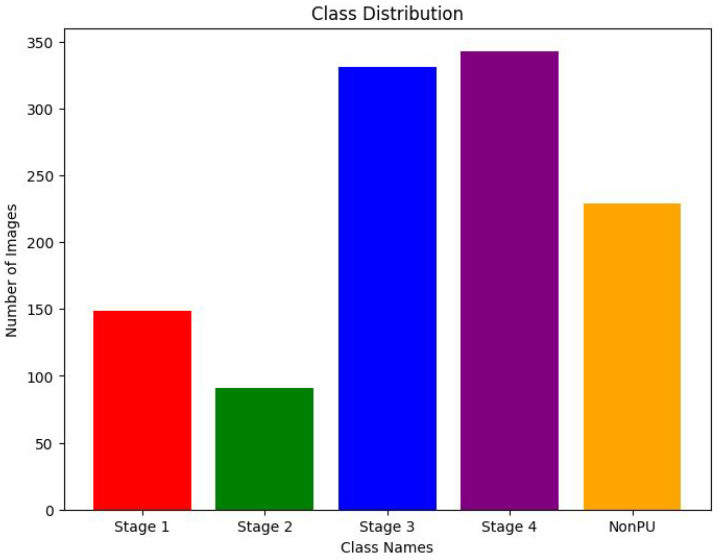
Bar chart showing the distribution of pressure ulcer stages and non-pressure ulcers in the dataset.

**Figure 6 healthcare-11-01222-f006:**
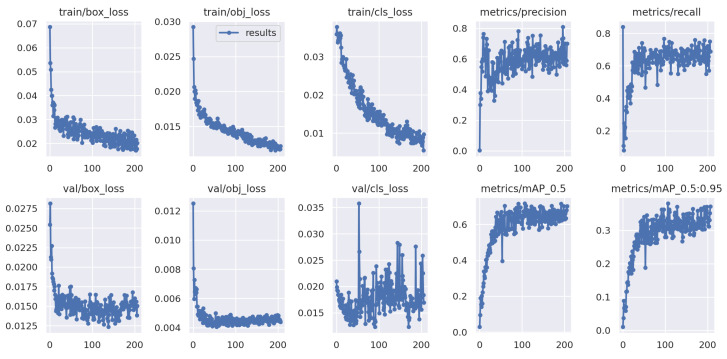
The training results.

**Figure 7 healthcare-11-01222-f007:**
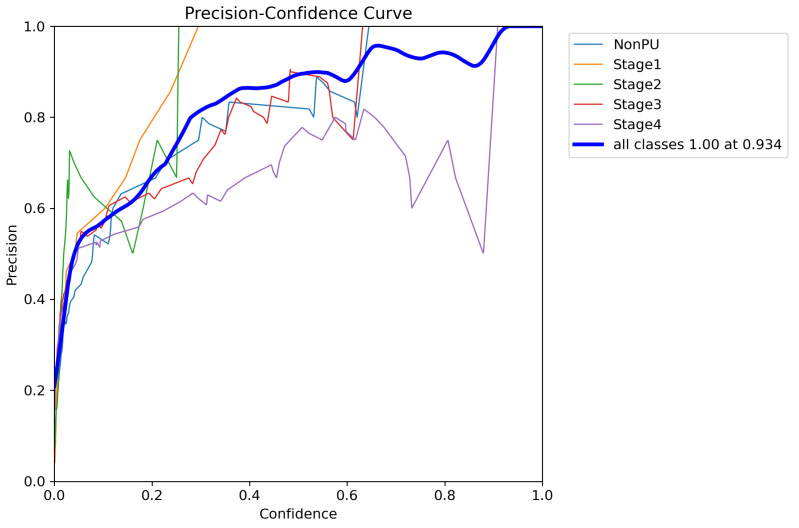
Precision confidence curve.

**Figure 8 healthcare-11-01222-f008:**
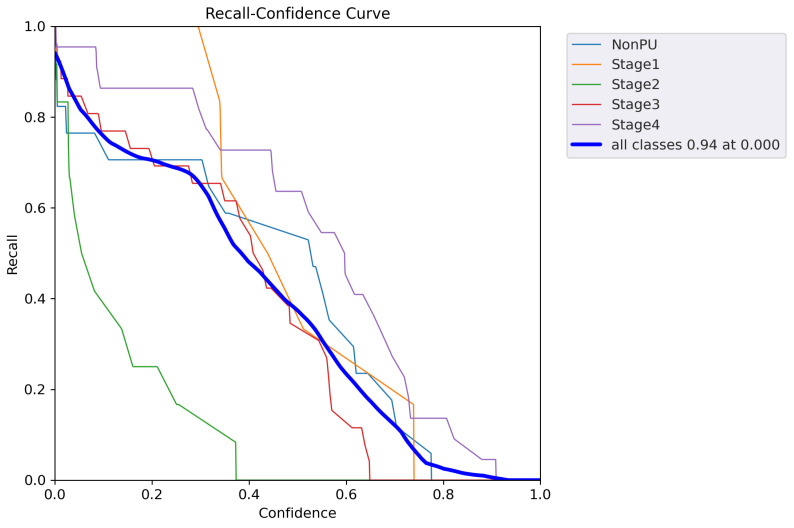
Recall confidence curve.

**Figure 9 healthcare-11-01222-f009:**
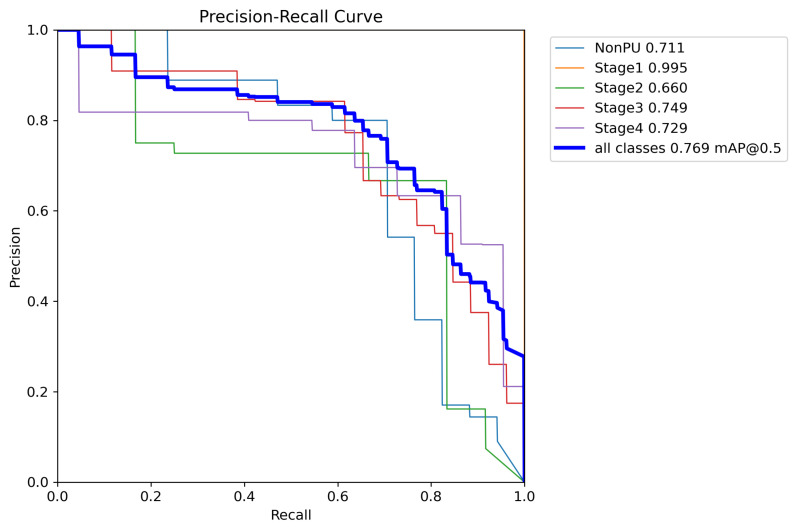
Precision–recall curve.

**Figure 10 healthcare-11-01222-f010:**
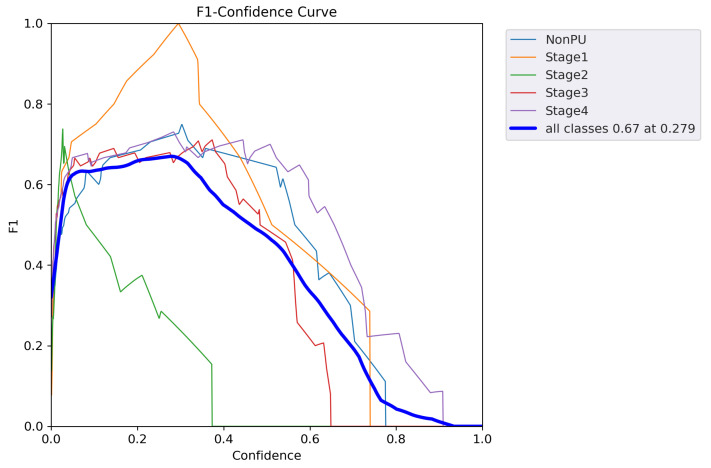
F1 score curve.

**Figure 11 healthcare-11-01222-f011:**
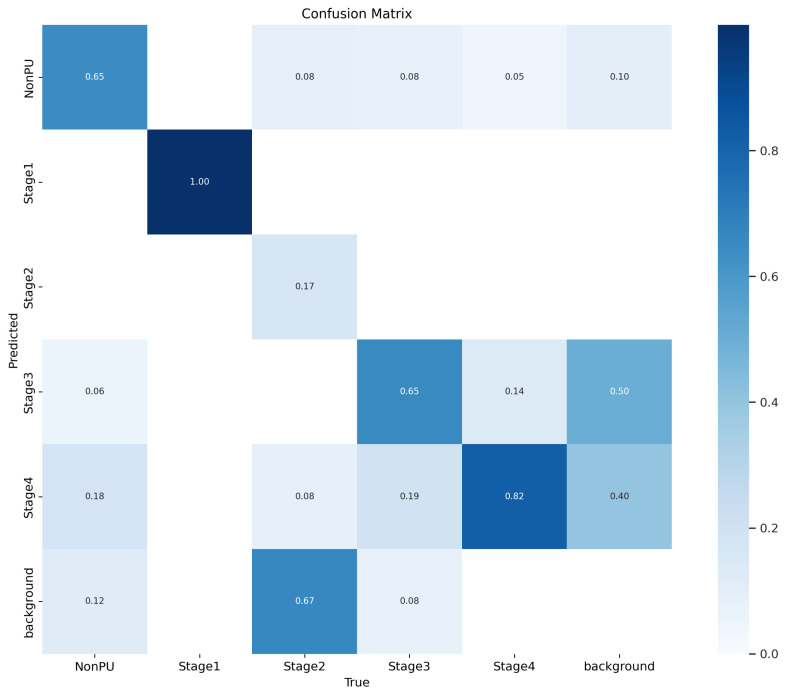
Confusion metrics.

**Table 1 healthcare-11-01222-t001:** The staging of pressure ulcers according to NPUAP.

Stage	Name	Description	General Features
1	Non-blanchable erythema	Skin is intact, but may appear red or discolored. It may feel warmer or cooler to the touch than the surrounding skin.	Discoloration, warmth or coolness, pain, itching
2	Partial-thickness skin loss	Skin or portion of skin that has lost some of its density appears as a small, open sore with no slough and a red or pink wound bed.	Abrasion, blistering, partial thickness loss, shallow open ulcer, red or pink wound bed
3	Full-thickness skin loss	Loss of tissue in its entirety, possible subcutaneous fat visibility, but no exposed bone, tendon, or muscle.	Deep open crater, full thickness loss, and visible subcutaneous fat, may have slough or necrotic tissue
4	Full-thickness skin loss with exposed bone, tendon, or muscle	A complete loss of tissue with exposed bone, muscle, or tendons.	Bone, tendon, or muscle that is visible whole thickness reduction, necrotic, or slough tissue

**Table 2 healthcare-11-01222-t002:** Computer vision detectable features for pressure ulcer stages.

Features	Stage 1	Stage 2	Stage 3	Stage 4
Color	Skin appears reddened	Pink wound in center and discoloration around the sore	Pus or a greenish fluid from the sore	Dark purple or black color in the area
Texture	Affected skin may be different from the surrounding healthy skin	The ulcer area may appear broken or damaged		The presence of eschar, firm, or mushy texture in the area
Border	The affected skin may have a clear border separating it from the surrounding healthy skin		Can involve undermining or tunneling, where the ulcer extends into the surrounding tissue	
Depth		Involves partial-thickness loss of skin	full-thickness tissue loss, which can be deeper than stage-2 ulcers	A total loss of tissue with exposed bone, muscle, or tendons

**Table 3 healthcare-11-01222-t003:** Brief overview of deep learning algorithms for item recognition and classification in medical images.

Study	Task	Method	Dataset	Limitations
[53]	Detection of lung nodules	Faster R-CNN	LIDC-IDRI	Does not include small benign nodules, smaller dataset
[54]	Detection and classification the breast tumors	Mask R-CNN	ultrasound images China Medical University Hospital	Small dataset size limit the generalizability
[55]	Brain tumor detection and classification using medical images	Hybrid DCNN with ResNet 152	Brats MRI image dataset	N/A
[56]	Fracture detection in wrist X-ray images	10 different object detection models including RetinaNet, Faster-RCNN, and RegNet	Clinical dataset collected from Gazi University Hospital	Single clinical dataset
[57]	Detect and classify adenomatous polyps.	Multi-branch CNN with attention module	6059 images	Not tested in clinical setting
[58]	Using colonoscopy images to detect and categorize colorectal polyps	CNN will enhance and filter the images, and a single shot multi-box detector (SSD) will identify and categorize any anomalies	27,508 whole slide images (WSI)	The dataset used is limited to images observed under white light imaging (WLI)
[59]	Detection and classification of breast cancer stages	CNN (Xception)	Collection of ultrasound images (USIs) of breast cancer patients	N/A

**Table 4 healthcare-11-01222-t004:** Summary of studies on pressure ulcer stages detection and classification using deep learning.

Study	Methodology	Results
[66]	Deep learning models and a LiDAR camera are used in a system for automatic segmentation and measuring	Achieved acceptable accuracy and a 26.2% mean relative error with U-Net outperforming Mask R-CNN
[67]	Employed CNNs for automatic tissue classification in pressure injuries	Achieved an overall classification accuracy of 92.01%, with high precision for granulation and necrotic tissue, and lower precision for slough tissue
[68]	Diagnosed pressure ulcers using deep learning algorithms	Achieved high accuracy, with the model classifying erythema and non-erythema wounds with an accuracy of about 98.5% and necrotic tissue with an accuracy of about 97%
[69]	Categorized and documented pressure ulcers using faster region-based CNN and mobile platform	Achieved 45 false positives with a mean average precision of 0.6796, recall of 0.6997, and F1 score of 0.6786 using a confidence score threshold of @.75.
[70]	Simultaneous segmentation and classification of pressure injuries using Mask-R-CNN algorithm	Achieved strong F1 scores and Dice coefficients for stages 1–4, resulting in a total classification accuracy of 92.6% and segmentation accuracy of 93.1%.
[71]	Tissue classification system for pressure ulcers using 3D CNN	Achieved an average of 95% AUC, 92% DSC, and 10% PAD.

**Table 5 healthcare-11-01222-t005:** Performance Evaluation of Pressure Ulcer Detection Model on Different Stages.

Class	P	R	mAP@50	mAP@50-95
all	0.781	0.685	0.769	0.398
NonPU	0.725	0.706	0.711	0.279
Stage 1	0.908	1	0.995	0.501
Stage 2	1	0.164	0.66	0.336
Stage 3	0.659	0.692	0.749	0.375
Stage 4	0.615	0.864	0.729	0.497

**Table 6 healthcare-11-01222-t006:** Comparison between our model and benchmark models.

Study	Model	Task	Images	Classes	mAP	Precision	Recall	F1
[68]	CNN	Erythema Classification	528	2	N/A	97.7%	95.5%	98.5%
[68]	CNN	Necrotic Tissue Classification	528	2	N/A	96.7%	96.7%	97%
[69]	Faster R-CNN with IOU@.50 and CS@.90	Pressure Ulcer Stage Classification	216	6	70.9%	77.4%	64%	69%
[29]	YOLOv4	Pressure Ulcer Stage Classification	216	6	63.2%%	N/A	N/A	N/A
Our Model	YOLOv5s	Pressure Ulcer Detection and Classification into four stages and Non-PU	1000+	5	76.9%	78.1%	68.5%	73.2%

## Data Availability

Not applicable.

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
