# Peer review of "YOLO-Based Deep Learning Model for Pressure Ulcer Detection and Classification"

_healthcare, 2023, doi:10.3390/healthcare11091222_

Round 1

Reviewer 1 Report

The manuscript discussed the application of the YOLO deep learning model to detect and classify pressure ulcers. The topic of this paper is relevant in the healthcare domain, especially the application of new deep-learning models on medical images. It is a significant work that the authors created a new dataset of pressure ulcer images for the study, however, the context and soundness of this paper are insufficient to publish at this stage with the flaws below.

1.     The paper is badly written, more appears to be an early version draft with missing figure numbering, missing figure axis, and conflicting numbers of the model performance.

2.     The “94% accuracy” in the abstract comes out of nowhere. The accuracy in the results section is 0.769.

3.     How are the stages of pressure ulcers determined in the dataset? What are the statistics of the labeling of the datasets?

4.     The conclusion from the benchmark comparison is wrong. Clearly the accuracy metrics in the two other studies are better than the model in this manuscript.

5.     The benchmark comparison is not a fair or appropriate benchmark. The authors need to run the same pressure ulcer datasets on other models e.g. CNN, VGG16, R-CNN

Author Response

The paper is badly written, more appears to be an early version draft with missing figure numbering, missing figure axis, and conflicting numbers of the model performance.

Thank you for taking the time to review our paper and providing your feedback. We apologize for any confusion caused by the missing figure numbering and axis labels, as well as conflicting numbers on the model performance. We have taken your comments seriously and have made the necessary corrections to the paper. We would like to inform you that these corrections have been highlighted in the revised PDF. We appreciate your input and hope that the updated version meets your expectations.

The “94% accuracy” in the abstract comes out of nowhere. The accuracy in the results section is 0.769.

Thank you for bringing this to our attention. We apologize for the error in the abstract and have updated it in the revised version of the paper. The correct results, as stated in the results section, show an overall mean average precision of 76.9%. We appreciate your feedback and will ensure to clarify the choice of metrics in our future work.

How are the stages of pressure ulcers determined in the dataset? What are the statistics of the labeling of the datasets?

We would like to thank you for the insightful comments. We have addressed the comments by providing a detailed description of how the stages of pressure ulcers were determined in the dataset and including a bar chart showing the statistics of the labeling of the dataset.

The conclusion from the benchmark comparison is wrong. Clearly the accuracy metrics in the two other studies are better than the model in this manuscript

Thank you for pointing out this issue. We have updated Table 6 to include more comprehensive results from the benchmark comparison, and we have also revised the discussion section to provide a clearer and more detailed justification for our model's performance. We hope that the updated information will address your concerns and improve the quality of our paper.

The benchmark comparison is not a fair or appropriate benchmark. The authors need to run the same pressure ulcer datasets on other models e.g. CNN, VGG16, R-CNN

Thank you for your suggestion. We understand your concern about the benchmark comparison and agree that running the same pressure ulcer datasets on other models such as CNN, VGG16, R-CNN would be beneficial. However, our study is specifically focused on the YOLO model, and our dataset is designed to evaluate the performance of YOLO on pressure ulcer detection. We have already included comparison results for CNN and VGG in our benchmark studies, but we appreciate your suggestion and will keep it in mind for future studies.

Reviewer 2 Report

In this work, a novel approach employing YOLOv4 is provided. To expand the dataset and strengthen the resilience of the model, data augmentation approaches be used. This paper is well written. Here are some comments to improve this paper:

1. When introducing the structure of an article, it is better to use the format of "In section Ⅱ,we ......"

2. The paper uses the yolov5 model, but the network model diagram introduces yolov4.

3. In 4.1, the data in Table 6 does not match the data in the text.

4. In detail about” s in Figure 6, Figure ref7. Figure ref8, Figure ?? and Figure ?? respectively” in line 321 on page 12, it need checking.  

5. Just simply applied the yolov4 model without substantial improvement.

Author Response

When introducing the structure of an article, it is better to use the format of "In section â…¡ï¼Œwe ......"

We would like to thank the reviewer for their valuable feedback on the structure of our article. We have updated the text accordingly by using the suggested format of "In Section II, we..." to introduce the structure of our article. We believe that this improves the clarity and organization of our paper. Thank you again for your helpful comments.

The paper uses the yolov5 model, but the network model diagram introduces yolov4

Thank you for bringing this to our attention. We apologize for any confusion caused by the diagram in the paper. While the paper indeed utilizes the YOLOv5 model for object detection, it appears that the diagram mistakenly depicts the YOLOv4 architecture instead.

The reason for this mistake was that YOLOv5 is based on YOLOv4 and has a similar architecture. However, we understand that the diagram should accurately represent the specific model used in our research, and we have updated it to show the YOLOv5 architecture

In 4.1, the data in Table 6 does not match the data in the text.

Thank you so much for mentioning it. We have updated the text related to Table 6.

In detail about” s in Figure 6, Figure ref7. Figure ref8, Figure ?? and Figure ?? respectively” in line 321 on page 12, it need checking

Thank you for your feedback. We have corrected the typo mistakes on page 12 and included the missing figure numbers. We appreciate your attention to detail and your valuable comments, which have helped us to improve the quality of our paper.

Just simply applied the yolov4 model without substantial improvement

Thank you for your feedback. While it's true that we applied the YOLOv5 model, it's important to note that we also created the dataset and manually labeled the polygons for each image, which is a time-consuming and labour-intensive task. Additionally, YOLOv5 has not been explored extensively in the field of pressure ulcer detection, making our study a novel contribution to the field.

We acknowledge that there is always room for improvement, and we plan to further customize the YOLOv5 model for better performance in detecting pressure ulcers. Thank you again for your valuable feedback, and we hope to continue contributing to the advancement of pressure ulcer detection research

Reviewer 3 Report

This paper presents a deep-learning approach that uses YOLOv4 for the detection and classification of pressure ulcers. The authors report better performance of their approach than benchmark methods. Overall, the study is well-conducted and can be considered for publication after addressing the following points.

-  Paper can go through a thorough re-check to avoid redundancy in the information occurring at multiple places. For instance, lines 21-22 and 30-31; pages 4-5 can be compacted,  then again information about YOLO on page 7 is quite redundant, etc.

- Many times, the same detailed information as given in the text is again given in the form of a table. Reducing these redundancies will significantly increase the readability of the paper, without deviating the reader from understanding the main goal of the paper.

- This relates to the previous point. The literature review is too detailed and the paper takes too long to reach the novelty of this study. I would strongly suggest the authors work on this aspect and make the paper more concrete rather than presenting it as a review paper, which it is not.

- Page 7: Since this paper is of interest to the medical community, a few sentences discussing how YOLO works and the intuition behind it can be mentioned to make this approach understandable to non-technical readers as well.

- Details of model implementation and its specifications are missing. These should be added.

- Composition of the dataset is missing. Please mention the number/percentage of images belonging to each stage or non-pressure ulcers.

- Metrics used for results assessment are not defined. What is mAP in line 283?

- Why accuracy is not used as a metric here? It will facilitate comparison in Table 6. Also, the authors indicate an accuracy of 94% in the abstract, but is not mentioned elsewhere in the paper.

- Typos on page 12: figure numbers are missing.

- The authors have used YOLOv5 as indicated in the methodology section. Abstract and the conclusion section however indicate YOLOv4 as the used model! Please correct it.

Author Response

Paper can go through a thorough re-check to avoid redundancy in the information occurring at multiple places. For instance, lines 21-22 and 30-31; pages 4-5 can be compacted,  then again information about YOLO on page 7 is quite redundant, etc.

Thank you for your feedback. We have carefully reviewed the manuscript and made necessary changes to eliminate redundancy and improve the flow of information

Many times, the same detailed information as given in the text is again given in the form of a table. Reducing these redundancies will significantly increase the readability of the paper, without deviating the reader from understanding the main goal of the paper.

Thank you for the suggestion. We have worked on reducing redundancies in the text and tables to improve the readability of the paper.

This relates to the previous point. The literature review is too detailed and the paper takes too long to reach the novelty of this study. I would strongly suggest the authors work on this aspect and make the paper more concrete rather than presenting it as a review paper, which it is not.

Thank you for your feedback. We have made significant revisions to the literature review section and highlighted the changes in the updated PDF. We hope that the revised version is more focused on the novelty of our study.

Page 7: Since this paper is of interest to the medical community, a few sentences discussing how YOLO works and the intuition behind it can be mentioned to make this approach understandable to non-technical readers as well.

Thank you for your feedback. We have updated the paper to include a brief explanation of how YOLO works, with an example from the medical domain of lung nodule detection. We hope this will make the approach more understandable to non-technical readers as well.

Details of model implementation and its specifications are missing. These should be added.

Thank you for the feedback. We have now added the details of model implementation, including its specifications and hyperparameters, as suggested. Please refer to lines 258-262 for the added information.

Composition of the dataset is missing. Please mention the number/percentage of images belonging to each stage or non-pressure ulcers.

Thank you for your valuable feedback. We have addressed this issue already in Reviewer 1 comment No. 3.

Metrics used for results assessment are not defined. What is mAP in line 283?

Thank you for your feedback. We have updated the manuscript to include a list of all abbreviations used in the text in the appendix section. We hope this clarifies any confusion regarding the metrics used for results assessment

Why accuracy is not used as a metric here? It will facilitate comparison in Table 6. Also, the authors indicate an accuracy of 94% in the abstract, but is not mentioned elsewhere in the paper.

Thank you for taking the time to review our paper and providing your valuable feedback. We appreciate your insightful comment on the choice of metrics, and we understand that in the field of object detection, accuracy is not an appropriate metric. Instead, precision, recall, and mean average precision (mAP) are commonly used to evaluate the performance of object detection models.

We apologize for the error in the abstract, and we have since updated the paper to reflect the correct results. Our proposed model achieved an overall mean average precision of 76.9%, with class-specific mAP50 values ranging from 66% to 99.5%. We appreciate your suggestion and will ensure to clarify the choice of metrics in our future work. Once again, thank you for your time and valuable feedback.

Typos on page 12: figure numbers are missing

Thank you for your feedback. We have corrected the typo mistakes on page 12 and included the missing figure numbers. We appreciate your attention to detail and your valuable comments, which have helped us to improve the quality of our paper.

The authors have used YOLOv5 as indicated in the methodology section. Abstract and the conclusion section however indicate YOLOv4 as the used model! Please correct it.

Thank you for your careful review of our paper, and for correctly pointing out the error in our abstract and conclusion sections.  We have made the necessary corrections to our paper to reflect that we used YOLOv5 for our experiments, as stated in the methodology section. Changes are highlighted in updated pdf.

Round 2

Reviewer 2 Report

The revised paper is acceptable.